# Generative Trajectory Planning in Dynamic Environments: A Joint Diffusion and Reinforcement Learning Framework

## Abstract

Real-time trajectory optimization requires planners that can simultaneously ensure safety and energy efficiency in environments containing both static and dynamic obstacles. This paper introduces a generalized framework that combines diffusion-based trajectory generation with deep reinforcement learning (DRL). The diffusion component generates diverse candidate trajectories by modeling feasible sub-paths, where a sub-path denotes a short-horizon segment aligned with receding-horizon execution. In this formulation, the entire trajectory is decomposed into consecutive sub-paths, enabling the diffusion model to learn local collision avoidance and smoothness while maintaining consistency across the fully identified path (e.g., global path and whole trajectory). The DRL component then evaluates these candidates online, selecting actions that improve safety while adapting to dynamic obstacles and maintaining energy-efficient behavior. The joint design leverages the generative diversity of diffusion and the adaptive decision-making of DRL, producing a planner that is both responsive and reliable. To assess effectiveness, the method is evaluated in unmanned aerial vehicle (UAV) path optimization scenarios with dynamic obstacles. The results demonstrate that sub-path training enhances the generalization of diffusion-based planners by linking local feasibility to global performance, and that the approach offers a practical solution for real-time UAV trajectory optimization with improved safety and efficiency.

## 1 Introduction

Trajectory generation in cluttered environments is a foundational capability for graphics, games, and autonomous driving. In these settings, the planner must produce short, smooth, and safe paths that consider static geometry while adapting to dynamic scene elements, all under tight latency budgets. A practical system must trade progress and distance against energy curvature and clearance, and it must do so quickly enough to close the loop as the environment changes. Classical sampling methods, such as $A^*$ and Rapidly-exploring Random Tree (RRT), and optimization methods offer guarantees in static worlds (LaValle, 1998; Hart et al., 1968). Their performance often deteriorates as dynamics and horizon length grow. Prediction followed by replanning becomes heavy, cost landscapes are brittle to tuning, and learned priors are awkward to inject. Recent generative approaches, especially diffusion models, encode compelling shape priors for trajectories (Wang et al., 2024b). Safety is then typically enforced with soft penalties, and inference time scales with the number of sampling steps. Reinforcement learning (RL) brings adaptivity, yet directly emitting full waypoint sequences, for example, three coordinates per step over a long horizon, is high-dimensional and hard to train stably, and it yields limited safety guarantees (Lillicrap et al., 2015).

These problems are addressed by replanning path generation as candidate selection. A diffusion-based path provider proposes a small, diverse set of $K$ candidates. These are built from line of sight primitives with lightweight warps and then refined by the generative prior. The learned policy network outputs a low-dimensional vector in $R^K$, which is mapped with a softmax and normalized over the subset of candidate trajectories that pass a safety filter to yield convex mixture weights. The mixed path is executed over a short horizon. If it violates static or dynamic constraints, the system immediately falls back to the maximum clearance candidate. Observations summarize the top M nearby dynamic objects using relative pose velocity radius and type with scale normalization for in-

variance. The policy is trained with Soft Actor Critic using a decomposed reward with explicit terms for progress distance, energy, curvature, and clearance. A lightweight Shield applies smoothing and a small lateral offset on detected grazes, which reduces jitter and near collisions without global replanning. This separation of concerns consists of three parts: priors for plausible paths, filters for immediate safety, and policies for adaptive trade-offs. These yield three practical benefits. First, it reduces control dimensionality, which stabilizes learning and clarifies credit assignment. Second, it exposes compute to quality knobs that map cleanly to latency. Third, it preserves interpretability because each reward term and each safety gate is tied to a specific, measurable behavior in the system, which makes tuning straightforward at deployment. In the proposed approach, the overall trajectory is further decomposed into multiple segments. For each segment, an optimal sub-path is identified by combining the generative capabilities of diffusion models with the adaptive decision-making of RL. This hierarchical formulation allows the planner to efficiently balance local safety with global progress in dynamic environments. The proposed framework is evaluated in a simulated UAV navigation environment. UAVs are chosen as the test domain because safe and efficient operation in cluttered 3D spaces requires continuous interaction with dynamic obstacles and real-time adaptation. This environment emphasizes the practical relevance of the proposed benefits, and the effectiveness of the approach is confirmed quantitatively through performance metrics.

## 2 RELATED WORK

### 2.1 DIFFUSION MODELS IN GENERATIVE MODELING

Diffusion models have recently attracted significant interest in the field of path planning and trajectory generation for autonomous driving. One line of work introduces a diffusion-based motion planning framework that learns a trajectory prior for robot motion and integrates it with a sampling-based planning algorithm, thereby achieving both collision avoidance and smooth path generation (Carvalho et al., 2023). Improving planning speed and efficiency has also been an important focus. For example, PRESTO employs a diffusion model conditioned on key configurations summarizing the environment to rapidly generate an initial trajectory, which is then refined through an optimization step to support real-time path planning in autonomous driving scenarios (Seo et al., 2025). Recent studies have further sought to bridge trajectory prediction and planning. MotionDiffuser, for instance, models the probabilistic distribution of future multi-agent trajectories with a diffusion process, and leverages a differentiable cost function to achieve controllable planning (Jiang et al., 2023). Along similar lines, optimization techniques for diffusion-based joint trajectory prediction have been shown to improve inference efficiency and controllability, resulting in stable planning performance even in complex road environments with multiple interacting agents (Wang et al., 2024b). Finally, transportation-focused research has explored intention-aware diffusion models, which incorporate the intent of pedestrians and vehicles into the trajectory generation process. This integration has been shown to enhance the reliability of both prediction and planning, and is increasingly recognized as a promising foundation for autonomous driving decision-making and safe operation (Liu et al., 2025).

### 2.2 DEEP REINFORCEMENT LEARNING FOR TRAJECTORY PLANNING

DRL has emerged as a central paradigm for trajectory planning in autonomous systems. Classical on-policy and off-policy methods remain influential. Proximal Policy Optimization (PPO) has been applied in autonomous driving, where human-in-the-loop corrections accelerate learning and improve safety (Shi et al., 2024). In continuous control domains, soft actor-critic (SAC) has been extended with risk modeling (Wang et al., 2024a). risk assessment-based SAC (RA-SAC) augments state features and reward design to generate smoother and more fuel-efficient routes. For high-dimensional tasks such as robotic manipulation, hierarchical reinforcement learning (HRL) decomposes planning into subproblems, improving robustness and sample efficiency (Bischoff et al., 2013). Recent work shifts the focus from single-step action selection toward sequence-level generation by integrating generative models into the control loop. Generative architectures have been used for trajectory prediction, such as TrajLearn for real-time flow forecasting and MTNet for travel-time and route estimation (Nadiri et al., 2025; Wang et al., 2022a). A more recent line of research leverages generative models as policy representations. Standard DRL policies often assume unimodal distributions (e.g., Gaussians), which are restrictive in multimodal decision problems (Dong et al., 2025).

Diffusion models, in contrast, naturally capture multimodal distributions, making them well-suited for trajectory-level control. Planning with Diffusion formulates trajectory optimization as an iterative denoising process, refining candidate paths (Janner et al., 2022). Extensions such as diffusion q-learning and maximum entropy diffusion policy further combine diffusion with RL objectives, replacing Gaussian parameterizations with diffusion-based ones to enable richer exploration while retaining maximum entropy guarantees (Wang et al., 2022b; Dong et al., 2025).

These advances mark a shift from reactive action mapping to generative trajectory policies. Building on this trend, the present work integrates diffusion-based sub-path generation with RL to enable real-time trajectory optimization that enhances safety and energy efficiency in dynamic environments.

### 2.3 DIFFUSION-BASED PLANNERS AND TRAJECTORY OPTIMIZATION

A number of diffusion-based planners have been proposed to generate feasible trajectories in static environments. These methods treat motion planning as a generative modeling problem and leverage diffusion processes to sample full trajectories or refine coarse initial guesses. Diffuser (Janner et al., 2022) formulates planning as iterative denoising in trajectory space, producing globally coherent paths but assuming a fixed static world without dynamic agents. Multi-path Diffusion (MPD) extends this idea by generating multiple diverse trajectory hypotheses and projecting each candidate onto obstacle-free regions through geometric consistency constraints, yet the projection operates only with static geometry. Minimum-Margin Diffusion (MMD) further encourages large-clearance trajectories by modifying the denoising objective to penalize proximity to static obstacles. Smooth-Margin Diffusion (SMD) incorporates curvature-based regularization to favor smooth trajectories, while still relying exclusively on static obstacle maps. Multi-Constraint Trajectory Diffusion (MCTD) adds multi-objective costs such as length, smoothness, and clearance, but its optimization is performed offline and does not address dynamic or time-varying obstacles. Finally, trajectory stitching approaches generate short local segments and concatenate them into a global trajectory, but the stitching is typically heuristic and does not incorporate dynamic interactions or feedback-based selection.

Overall, these diffusion-based planners provide strong priors for feasible motion in static scenes but operate in an open-loop manner and assume that the entire trajectory can be generated before execution. They do not model dynamic obstacles, do not incorporate reactive decision making, and lack mechanisms to adjust to time-varying environments. In contrast, our framework integrates sub-path diffusion with online reinforcement learning, enabling closed-loop adaptation, dynamic obstacle avoidance, and real-time replanning under uncertainty.

## 3 PRELIMINARIES

### 3.1 OVERVIEW OF DIFFUSION MODELS

Diffusion models learn to invert a forward noising process using a neural denoiser. The forward process with variance schedule $\beta_t$ is

$$x_t = \sqrt{\bar{\alpha}_t}\, x_0 + \sqrt{1 - \bar{\alpha}_t}\, \varepsilon, \quad `\varepsilon \sim \mathcal{N}(0, I), \tag{1}$$

where $x_0$ denotes the original data, $\varepsilon$ is Gaussian noise, $\alpha_t = 1 - \beta_t$ is the retained signal coefficient at step $t$, and $\bar{\alpha}_t = \prod_{s=1}^{t} \alpha_s$ is the cumulative product controlling how much of the original data survives after $t$ noising steps. The reverse denoising distribution is parameterized as,

$$p_\theta(x_{t-1} \mid x_t) = \mathcal{N}\big(x_{t-1} \mid \mu_\theta(x_t, t),\, \Sigma_\theta(x_t, t)\big), \tag{2}$$

where $\mu_\theta$ and $\Sigma_\theta$ are outputs of a neural network with parameters $\theta$ that approximate the true mean and variance of the reverse conditional. Intuitively, this network predicts how to remove one step of noise from $x_t$ to recover a cleaner $x + t - 1$. Classifier-free guidance introduces a mechanism for controlling how strongly the generation follows the conditioning signal $c$. It interpolates between unconditional and conditional predictions of the noise as,

$$\hat{\varepsilon}_\theta(x_t, t, c) = \varepsilon_\theta(x_t, t, \emptyset) + w\Big(\varepsilon_\theta(x_t, t, c) - \varepsilon_\theta(x_t, t, \emptyset)\Big), \tag{3}$$

where $\varepsilon_\theta(x_t, t, \emptyset)$ is the unconditional prediction, $\varepsilon_\theta(x_t, t, c)$ is the conditional prediction, and $w > 1$ is the guidance scale that balances fidelity against diversity. Larger $w$ values force the model to adhere more strongly to $c$, often at the cost of reduced diversity.

## 3.2 Maximum Entropy Deep Reinforcement Learning

Diffusion priors provide trajectory level structure and the candidate-selection framework reduces control dimensionality, yet an additional mechanism is required to avoid premature convergence to a narrow subset of behaviors during online operation. In dynamic environments with shifting constraints, purely reward-maximizing policies often under-explore the action space, leading to fragile decision making and training instability. Maximum entropy reinforcement learning addresses this issue by introducing an entropy regularization term that discourages the over-concentration of probability mass on a small set of actions and maintains exploratory behavior throughout training. Unlike classical objectives that solely maximize expected return and are prone to insufficient exploration and local optima, the entropy-regularized formulation explicitly trades off task reward against policy randomness. This regularization is particularly well-suited for the candidate-selection setting, where the policy repeatedly chooses among a feasible set of diffusion generated sub-paths under dynamic, time-varying conditions. By sustaining stochasticity, the policy avoids premature collapse to suboptimal preferences.

Formally, the maximum entropy objective augments the standard expected return with an entropy bonus that encourages exploration as,

$$J_{\text{MaxEnt}}(\pi) = \mathbb{E}_{\tau \sim \pi}\left[ \sum_{t=0}^{\infty} \gamma^t \Big( R(S(t), A(t)) + \alpha \mathcal{H}(\pi(\cdot|S(t))) \Big) \right], \tag{4}$$

where $S(t) \in \mathcal{S}$ is the state, $A(t) \in \mathcal{A}$ the action, $R(S(t), A(t))$ the reward at step $t$, and $\gamma \in [0, 1)$ the discount factor. Then, the entropy of the policy can be as,

$$\mathcal{H}(\pi(\cdot|s)) = -\mathbb{E}_{a \sim \pi(\cdot|s)}[\log \pi(a|s)], \tag{5}$$

and the temperature parameter $\alpha > 0$ controls the trade-off between maximizing reward and encouraging randomness. A larger $\alpha$ promotes exploration by favoring higher entropy. The optimal value functions under this objective take the following soft forms,

$$V^*(s) = \alpha \log \int_{\mathcal{A}} \exp\Big( \tfrac{1}{\alpha} Q^*(s, a) \Big) \, da, \tag{6}$$

$$\pi^*(a \mid s) = \exp\Big( \tfrac{1}{\alpha} \big( Q^*(s, a) - V^*(s) \big) \Big), \tag{7}$$

where $Q^*(s, a)$ is the optimal soft state-action value function and $V^*(s)$ is the corresponding soft value. Intuitively, the optimal soft value $V^*(s)$ aggregates all possible actions at state $s$ using a soft maximum rather than a hard maximum, while the optimal policy $\pi^*(a|s)$ assigns higher probability to actions with larger $Q^*(s, a)$ but retains nonzero probability mass on suboptimal actions to encourage exploration. This formulation produces policies that are simultaneously reward-seeking and entropy-seeking, leading to behaviors and stable learning.

# 4 Proposed Method

## 4.1 System Model and Architecture

Diffusion models have difficulty guaranteeing real-time performance and stability, while RL on its own struggles with an excessively large continuous action space to explore. To overcome these limitations, a hybrid framework is proposed that integrates a generative diffusion model with an RL-based selector, as illustrated in Fig. 1. The diffusion module first generates a diverse set of candidate sub-paths, each annotated with attributes such as uncertainty, collision probability, and length. The environment provides state information, including the current position, goal, and obstacles, which are combined with candidate features to form a state representation. At each planning step $t$, the policy observes this representation and selects the optimal trajectory from the candidate set based on a multi-objective reward function. Rather than producing an entire trajectory at once, the process is executed in a receding-horizon manner, where short sub-paths are repeatedly proposed, evaluated, and executed. The design improves adaptability to dynamic environments and balances multiple reward components such as goal reaching, collision avoidance, energy efficiency, smoothness, and clearance. The state representation is defined as

$$S(t) = \big[ \Delta g_t, d_t, c_t, b_t, \ell^{(1:K)}, f^{(1:K)}, \bar{v}_t \big], \tag{8}$$

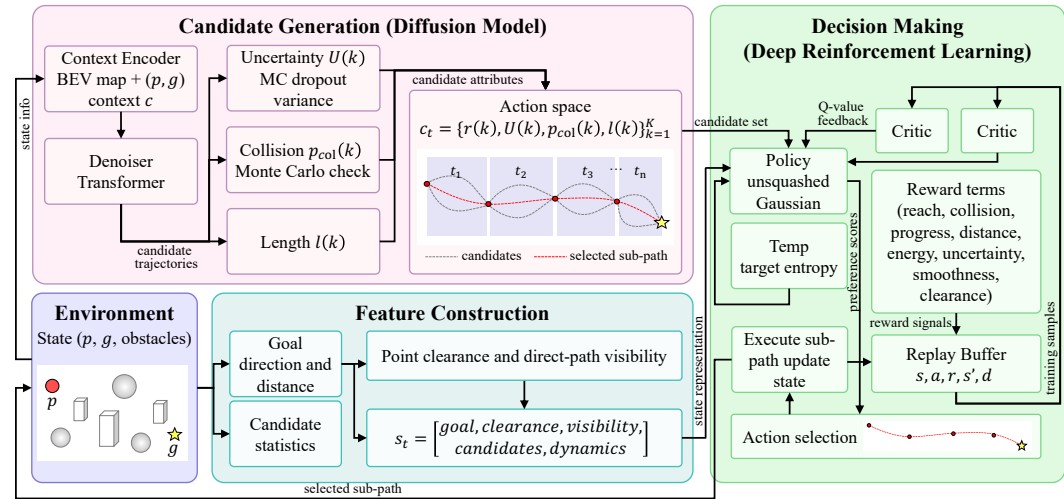

Figure 1: The architecture of the proposed diffusion-RL framework for trajectory generation. The environment provides state information, a diffusion module proposes candidate sub-paths, and an RL policy selects and executes safe trajectories in a receding-horizon loop.

where $\Delta g_t$ is the relative displacement to the goal, $d_t$ is the goal distance, $c_t$ is the clearance to obstacles, $b_t$ is a binary line-of-sight indicator, $\ell^k$ and $f^{(k)}$ denote length and feasibility of the $k$-th candidate, and $\bar{v}_t$ represents average environmental dynamics.

As demonstrated in Fig. 1, the framework consists of four main components environment, candidate generation using diffusion model, feature construction, and decision making with deep reinforcement learning. The overall flow begins with the environment which provides state information such as current position, goal and obstacles. This information is used by candidate generation to propose diverse sub path trajectories with attributes including uncertainty collision probability and length. Feature construction then encodes geometric and dynamic statistics by combining candidate information with goal direction clearance and visibility. Finally decision making with reinforcement learning evaluates these features and selects the most appropriate trajectory, balancing safety efficiency and adaptability in dynamic environments.

## 4.2 SUB-PATH DIFFUSION FOR CANDIDATE GENERATION

To provide flexibility under changing conditions, the overall trajectory is decomposed into fixed-length sub-paths. Unlike conventional diffusion planners that attempt to generate the entire trajectory at once, the proposed framework restricts prediction to a short rolling horizon. As illustrated in the candidate generation part of the action space in Fig. 1, at each step the diffusion model generates candidate sub-paths only within the upcoming horizon window, while the RL policy selects one sub-path to execute. When the agent moves forward, the horizon shifts and the next short candidate set is generated, ensuring that the prediction range remains consistently limited. This rolling-horizon design reduces computational cost, improves responsiveness to dynamic obstacles, and mitigates error accumulation over long horizons. Formally, the diffusion model outputs a candidate set at each step,

$$\mathcal{C}_t = \{\tau^{(k)}, U^{(k)}, p_{\text{col}}^{(k)}, \ell^{(k)}\}_{k=1}^K, \qquad (9)$$

where $\tau^{(k)}$ is the $k$-th sub-path, $U^{(k)}$ is its uncertainty score derived from variance estimates, $p_{\text{col}}^{(k)}$ is the estimated collision probability, and $\ell^{(k)}$ is its length. This representation provides both diversity and reliability, ensuring that downstream selection has sufficient options while maintaining safety-awareness. From a computational perspective, generating a full trajectory of horizon length $T$ requires a complexity on the order of $\mathcal{O}(M \cdot T \cdot H)$, where $M$ is the number of denoising steps and $H$ is the number of replanning iterations. In contrast, when decomposed into sub-paths of length

$T_{\text{sub}} \ll T$, the complexity reduces to $\mathcal{O}(M \cdot T_{\text{sub}} \cdot H)$. The relative reduction is expressed as

$$\frac{\text{Cost}_{\text{sub}}}{\text{Cost}_{\text{full}}} = \frac{T_{\text{sub}}}{T}. \tag{10}$$

Thus, shorter sub-paths yield a proportional speedup in candidate generation. Beyond computational efficiency, sub-path decomposition enhances adaptability in dynamic environments. Since only the upcoming segment needs to be re-generated when new obstacles appear, replanning latency is reduced, and the system remains responsive to unexpected changes. In this way, sub-path diffusion not only reduces complexity but also enhances robustness by enabling rapid local corrections without discarding the entire trajectory. Segmenting a path into sub-paths allows for the recalculation of only a portion of the trajectory when environmental changes occur, rather than discarding the entire path. This approach leverages the diversity provided by diffusion models to ensure fallback options even in out-of-distribution scenarios, thereby enhancing adaptability.

### 4.3 REINFORCEMENT LEARNING-BASED PATH SELECTION

Existing selection methods provide only short-term improvements and cannot capture long-horizon trade-offs between safety, efficiency, and smoothness. To overcome these limitations, the selection process is formulated as a Markov Decision Process (MDP) and optimized using RL, which enables principled exploration and stable learning. Given the candidate set $\mathcal{C}_t$ generated by the diffusion model, the system requires a principled mechanism for selecting the most appropriate trajectory. This selection process is formulated as a MDP, defined by the tuple $\mathcal{M} = \{\mathcal{S}, \mathcal{A}, \mathcal{P}, r, \gamma\}$, where $\mathcal{S}$ is the state space, $\mathcal{A}$ the action space, $\mathcal{P}$ the transition dynamics, $r$ the reward function, and $\gamma \in (0, 1)$ the discount factor. At the replanning step $t$, the environment provides a state vector

$$S(t) = \left[g - p_t, |g - p_t|, d_{\min}(p_t), b_t, \ell^{(1:K)}, f^{(1:K)}, \bar{v}_t\right], \tag{11}$$

which encodes the relative goal direction and distance, the clearance to obstacles, a binary line-of-sight indicator, the lengths of all candidate sub-paths, pre-check feasibility flags, and the average speed of dynamic obstacles. The SAC policy outputs a $K$-dimensional score vector

$$A(t) \in \mathbb{R}^K. \tag{12}$$

The environment interprets this vector as candidate preferences and maps it to a discrete choice among the $K$ diffusion-generated sub-paths. Executing action $A(t)$ therefore corresponds to following the selected candidate trajectory $\tau^{(k)}$ for the next motion segment. The decision policy is defined through a Q-function that estimates the expected cumulative return of selecting action $a$ in state $s$.

$$Q(s, a) = \mathbb{E}\left[\sum_{t=0}^{\infty} \gamma^t R(S(t), A(t)) \Big| s_0 = s, a_0 = a\right]. \tag{13}$$

The decision rule is

$$a^* = \arg\max_{a \in \{1, \ldots, K\}} Q(S(t), a), \tag{14}$$

ensuring that at each replanning step, the candidate with the highest long-term utility is chosen. The reward function $R(t)$ integrates multiple interpretable objectives, reflecting both task-oriented and safety-oriented considerations.

$$R(t) = w_g r_g(t) + w_c r_c(t) + w_p r_p(t) + w_d r_d(t) + w_u r_u(t) + w_v r_v(t) + w_l r_l(t). \tag{15}$$

where $r_g(t)$ and $r_c(t)$ represent goal-reaching and collision outcomes, $r_p(t)$ and $r_d(t)$ capture goal progress and residual distance, $r_u(t)$ penalizes uncertainty, $r_v(t)$ discourages excessive curvature, and $r_l(t)$ rewards clearance from obstacles. The weights $\{w_g, \ldots, w_l\}$ control the trade-offs between safety, efficiency, and stability. This integration of diffusion-based candidate generation and RL-based selection yields two key benefits. First, the diffusion model supplies a diverse and safety-aware candidate set at each step. Second, RL provides a principled selection mechanism that balances immediate feasibility with long-term efficiency. Moreover, by retaining the continuous policy formulation of SAC, entropy regularization can be directly applied, maintaining a balance between exploration and exploitation. This prevents premature convergence to suboptimal and enhances robustness under uncertainty, leading to more stable behavior in dynamic environments.

# 5 EXPERIMENTS

## 5.1 EXPERIMENTAL SETUP

Evaluation is conducted in a simulated 3D UAV planning environment containing both static and dynamic obstacles. UAV autonomous navigation typically involves interaction with dynamic entities such as vehicles, pedestrians, birds, or other UAVs. Accordingly, experiments in this setting allow for a realistic evaluation of the proposed algorithm under conditions where rapid and reliable responses to moving obstacles are essential. Safe and efficient navigation requires real-time adaptation to time-varying constraints, which makes UAV environments suitable for assessing the effectiveness of the proposed diffusion–RL framework.

Static obstacles are represented by randomly placed axis-aligned bounding boxes and spheres, while dynamic obstacles are modeled as agents that move with stochastic velocities and reflect upon world boundaries. Each episode begins with the UAV initialized at a random start position, and the target goal is sampled within a bounded region.

### 5.1.1 MODELING

To reflect UAV-specific considerations, the environment incorporates simplified kinematic motion and an energy consumption model. The UAV state evolves in discrete steps, advancing along sub-paths of fixed temporal length. The simulator evaluates multiple aspects of each executed trajectory: the path length $L(\tau)$ and aggregate curvature $\kappa(\tau)$ capture maneuver efficiency, while the minimum clearance $d_{\min}(\tau)$ quantifies safety with respect to obstacles. Energy consumption $E(\tau)$ is accumulated from both propulsion dynamics and onboard compute operations. In addition, success ratio, collision ratio, and time-to-goal are tracked to measure overall task performance. The detailed algorithm and hyperparameters are provided in A.

### 5.1.2 BASELINES

To demonstrate the effectiveness of the proposed diffusion-RL architecture, three representative classes of baselines are considered.

- *Heuristic planning:* Straight-line navigation and greedy waypoint tracking are used as simple heuristics. The straight-line planner directly connects start and goal without obstacle awareness, while greedy tracking iteratively steers toward the goal while avoiding collisions locally.

- *Classical sampling-based planners:* Standard motion planning algorithms such as RRT are included. These approaches explore the search space to construct feasible paths, but lack the learned adaptability of data-driven methods.

- *Learning-based trajectory generation:* A pure diffusion model without RL guidance is used to generate candidate sub-paths. Candidate selection is performed by random sampling or simple scoring heuristics, allowing us to evaluate the added value of RL in the online selection loop.

### 5.1.3 METRICS

Performance is assessed across both safety and efficiency dimensions. The primary performance metrics are success rate, collision rate, energy consumption, path efficiency, clearance, and uncertainty. The success rate measures the fraction of episodes where the UAV reaches its goal, while the collision rate measures failures due to static or dynamic obstacles. Energy consumption evaluates propulsion and computes costs normalized by trajectory duration. Path efficiency is calculated as the ratio between the shortest feasible distance and the actual path length. Clearance reflects the minimum distance to obstacles along the trajectory, and the uncertainty score is derived from diffusion variance, reflecting robustness to sampling variability. Together, these metrics provide a comprehensive evaluation of safety, efficiency, and reliability for UAV trajectory generation.

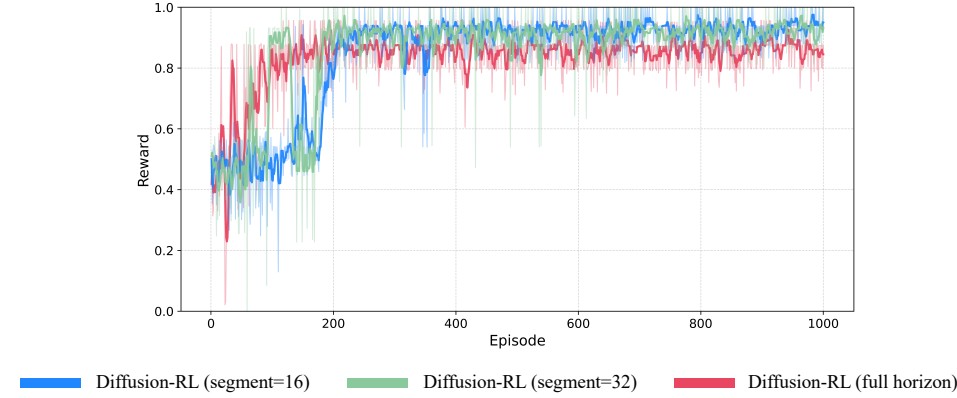

Figure 2: Training rewards versus episode. The full-horizon model learns fastest initially but is overtaken by the segmented variants (segment=16, segment=32), which converge to higher returns. Segment 16 is the most stable and attains the highest asymptotic reward.

## 5.2 RESULTS AND DISCUSSIONS

As shown in Fig. 2, the full-horizon model attains higher returns at the very beginning of training, indicating faster early learning. However, after about 200 episodes, both segmented variants (segment=16 and segment=32) consistently overtake the full-horizon baseline and converge to higher return levels. This pattern implies that segment-wise decision making stabilizes optimization and improves final policy quality, whereas the full-horizon setting exhibits larger variance and weaker convergence. Among the segmented models, segment=16 shows the most stable trajectory and the highest asymptotic return. Table 1 presents the evaluation results of the proposed diffusion-

Table 1: Comparison of planning strategies

| Method | Success rate | Collision rate | Timeout rate | Mean energy consumption | Mean obstacle clearance |
|---|---|---|---|---|---|
| **Diffusion-RL (segment=16)** | 0.92 | 0.00 | 0.08 | 348.8 | 58.20 |
| **Diffusion-RL (segment=32)** | 1.00 | 0.00 | 0.00 | 392.9 | 59.20 |
| Diffusion-RL (full horizon) | 0.52 | 0.00 | 0.48 | 168.9 | 50.70 |
| Heuristic | 0.32 | 0.68 | 0.00 | 1065.8 | 5.55 |
| Classical planner | 0.37 | 0.63 | 0.00 | 505.1 | 4.90 |
| Diffusion model | 0.27 | 0.01 | 0.72 | 406.1 | 53.00 |

RL framework under different segment lengths (16, 32, and full horizon), compared to representative baselines including heuristic planners, classical sampling-based methods, and a pure diffusion model. All models are tested under identical static and dynamic obstacle environments. The results highlight several findings. First, heuristic planners achieved low success rates while consuming excessive energy and frequently colliding with obstacles, indicating that simple rule-based navigation is inadequate for complex environments. Classical planners such as RRT performed somewhat better, but still suffered from low success rates and high collision rates, reflecting their limited adaptability in dynamic scenarios. Second, the pure diffusion model without RL guidance produced feasible trajectories but lacked robustness, further underscoring the importance of adaptive candidate selection. In contrast, the proposed diffusion-RL approach with segment planning achieved markedly superior performance. Both the 16- and 32-step configurations attained near-perfect success rates with zero collisions, while maintaining lower energy consumption and on average larger clearance from obstacles. This demonstrates the advantage of combining diffusion-based proposals with RL-based online decision making. Notably, the 32-step variant delivered the most consistent balance between success rate, safety, and efficiency. On the other hand, the full-horizon variant showed reduced reliability, suggesting that overly long planning horizons are less effective in dynamic environments where frequent replanning is required. These findings confirm that diffusion-RL with appropriate

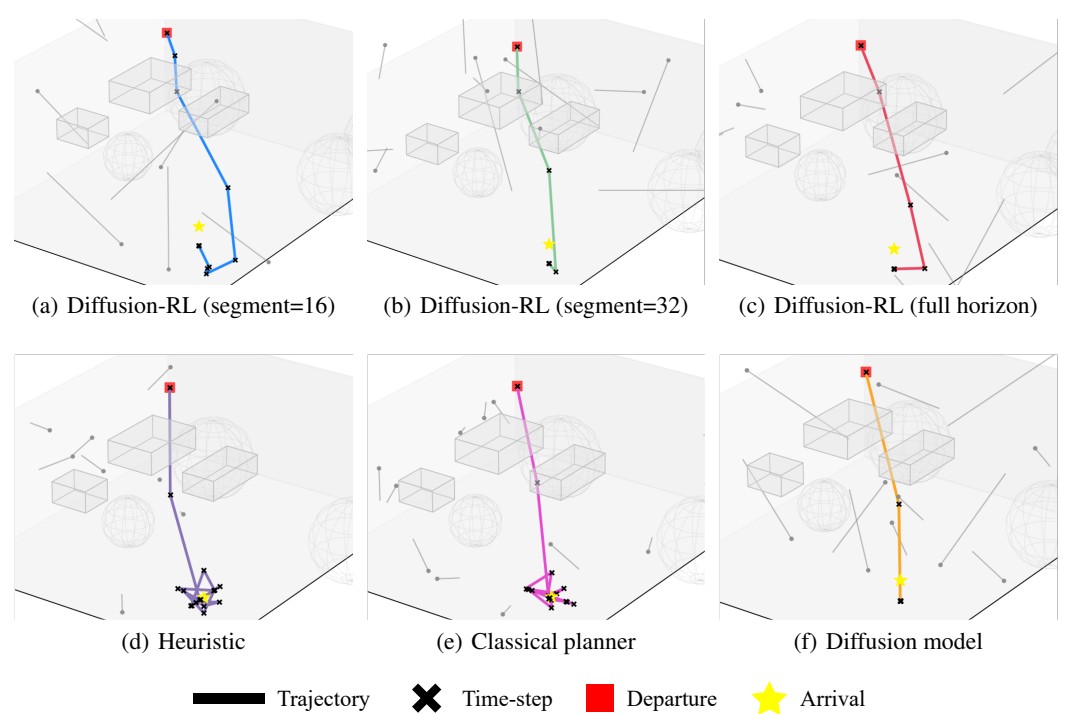

(a) Diffusion-RL (segment=16)   (b) Diffusion-RL (segment=32)   (c) Diffusion-RL (full horizon)

(d) Heuristic   (e) Classical planner   (f) Diffusion model

━━━ Trajectory   ✖ Time-step   ■ Departure   ★ Arrival

Figure 3: Trajectories generated by 6 planners in a 3D scenario.

segmentation outperforms traditional baselines and pure generative models, offering both robustness and efficiency in challenging navigation tasks.

As illustrated in Fig. 3, the qualitative trajectories further validate these quantitative results. Heuristic and classical planners often produce jagged or inefficient paths, sometimes failing to reach the goal when obstacles appear in close proximity. The pure diffusion model generates smoother paths but occasionally drifts toward unsafe regions due to the lack of adaptive selection. In contrast, the diffusion-RL variants yield trajectories that are both smooth and collision-free, adapting effectively to dynamic obstacles by continuously replanning sub-paths. Notably, the segmented approaches maintain shorter and safer prediction horizons, which results in tighter clearance around obstacles and more reliable progress toward the goal. These observations confirm that horizon segmentation not only stabilizes policy training but also translates into safe behaviors at execution time. This balance highlights that the choice of segment length serves as a practical design parameter to adjust the trade-off between stability and efficiency depending on deployment requirements.

More broadly, the results illustrate the complementary roles of diffusion and reinforcement learning within the proposed framework. Diffusion provides a diverse and semantically meaningful set of candidate sub-paths, while RL adaptively selects the most appropriate one given the current environment and task objectives. This division of labor combines the generative strengths of diffusion with the decision-making adaptivity of RL, producing trajectories that are both feasible and context-aware. Such a synergy suggests that diffusion-RL integration can serve as a promising paradigm for trajectory planning in complex and dynamic environments.

## 6 CONCLUSION

This paper introduces a framework that fuses diffusion-based planning with DRL to produce safe, energy-efficient 3D trajectories amid static and dynamic obstacles. Long horizons are split into fixed-length sub-paths, with a diffusion model that proposes diverse, safety-aware candidates, and RL selects among them online. This design reduces computational load, improves responsiveness, and mitigates long-horizon error. In UAV simulations, the approach outperforms heuristic planners, classical planners, and pure diffusion, with segment-wise selection yielding high success,

near-collision-free execution, strong clearance, and low energy use. These results show diffusion and RL are complementary for real-time planning under uncertainty and indicate applicability to autonomous driving, multi-robot coordination, and manipulation. Future work includes scaling to higher-dimensional systems, integrating semantic information into candidate generation, and systematically evaluating transfer to real-world robotic platforms. In addition, promising directions include developing theoretical guarantees for safety and stability under diffusion-based candidate generation, reducing computational overhead through lightweight diffusion sampling techniques, and integrating multi-modal sensory cues such as vision and language to support more informed decision making. Another avenue is benchmarking across standardized large-scale simulation suites and deploying on physical platforms, which will be essential steps toward validating the practicality and robustness of the framework in diverse, real-world scenarios.

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

# A APPENDIX

PROOF OF DIFFUSION AND RL ALGORITHM

A diffusion model learns to reverse a fixed forward noising process. Given clean data $x_0$, the forward process is

$$q(x_t|x_0) = \sqrt{\bar{\alpha}_t}\, x_0 + \sqrt{1 - \bar{\alpha}_t}\, \epsilon, \qquad \epsilon \sim \mathcal{N}(0, I), \tag{16}$$

where $\alpha_t = 1 - \beta_t$, $\bar{\alpha}_t = \prod_{i=1}^{t} \alpha_i$, and $\{\beta_t\}$ is a variance schedule.

A neural denoiser $\varepsilon_\psi(x_t, t, c)$ is trained to predict $\epsilon$ (or $x_0$ or $v$, depending on parameterization). The standard training objective is a mean-squared error:

$$\mathcal{L}_{\text{DM}} = \mathbb{E}_{t, x_0, \epsilon}\big[\|\epsilon - \varepsilon_\psi(x_t, t, c)\|_2^2\big]. \tag{17}$$

DDIM (denoising diffusion implicit models) provides a deterministic or low-variance sampling rule:

$$x_{t-1} = \sqrt{\bar{\alpha}_{t-1}}\, \hat{x}_0 + \sqrt{1 - \bar{\alpha}_{t-1}}\, \hat{\epsilon}, \tag{18}$$

where $\hat{x}_0$ and $\hat{\epsilon}$ are obtained from the network prediction. Optional noise $\sigma_t z$, $z \sim \mathcal{N}(0, I)$ can be added for stochastic sampling.

**Classifier-free guidance** To amplify conditional signals, one can interpolate between conditional and unconditional predictions:

$$\widehat{g}_\psi(x_t, t, c) = \hat{g}_\psi(x_t, t, \emptyset) + s\big(\hat{g}_\psi(x_t, t, c) - \hat{g}_\psi(x_t, t, \emptyset)\big), \quad s \geq 1. \tag{19}$$

At each time step, the agent observes $S(t)$, samples an action $A(t) \sim \pi_\theta(\cdot|S(t))$, receives reward $R(t)$, and transitions to $s_{t+1}$.

**Soft value functions** Define the soft $Q$-function and state value:

$$Q^\pi(S(t), A(t)) = R(t) + \gamma\, \mathbb{E}_{s_{t+1}, a_{t+1} \sim \pi}\big[V^\pi(s_{t+1})\big], \tag{20}$$

$$V^\pi(S(t)) = \mathbb{E}_{A(t) \sim \pi}\big[Q^\pi(S(t), A(t)) - \alpha \log \pi(A(t)|S(t))\big], \tag{21}$$

where $\alpha$ is the temperature controlling entropy weight.

**Soft Bellman backup** The target for critic learning is

$$y = r + \gamma\left(\min_j Q_{\phi'_j}(s', a') - \alpha \log \pi_\theta(a'|s')\right), \quad a' \sim \pi_\theta(\cdot|s'). \tag{22}$$

**Policy update** The actor is trained by minimizing

$$J_\pi(\theta) = \mathbb{E}_{s \sim \mathcal{D}, a \sim \pi_\theta}\big[\alpha \log \pi_\theta(a|s) - \min_j Q_{\phi_j}(s, a)\big], \tag{23}$$

where $\mathcal{D}$ is the replay buffer.

**Temperature update** The temperature $\alpha$ is adapted to match a target entropy $\mathcal{H}^\star$:

$$J(\alpha) = \mathbb{E}_{a \sim \pi_\theta}\big[-\alpha\big(\log \pi_\theta(a|s) + \mathcal{H}^\star\big)\big]. \tag{24}$$

Gradient descent on $J(\alpha)$ adjusts $\alpha$ automatically.

ALGORITHM

---

**Algorithm 1** Integrated Training and Inference of Diffusion-RL Framework

---

**Require:** Dataset $\mathcal{D}$, diffusion steps $T$, segment length $L$, DRL hyperparameters $(\alpha, \gamma, \tau, \eta)$

**Ensure:** Trained diffusion components $(E_\phi, f_\theta)$ and DRL policy $\pi_\psi$ with critics $Q_{\theta_1}, Q_{\theta_2}$

  **Diffusion model** Condition encoder $E_\phi$ maps state, goal, and obstacle map to a compact context $c$. The denoiser $f_\theta$ is a Transformer-based network that predicts noise or clean trajectories at each diffusion step.

  **DRL agent** The policy $\pi_\psi$ outputs a distribution over candidate sub-paths, while twin critics $Q_{\theta_1}, Q_{\theta_2}$ estimate long-term returns. Target networks are used for stability.

  **Training loop**

  **for** each epoch **do**

   **for** mini-batch $(\tau, s, g, \mathcal{O}) \sim \mathcal{D}$ **do**

    Generate sub-path segment $x_0$ (length $L$) with anchored endpoints.

    Encode condition $c \leftarrow E_\phi(s, g, \mathcal{O})$.

    Apply forward diffusion $x_t = \sqrt{\bar{\alpha}_t} x_0 + \sqrt{1 - \bar{\alpha}_t}\epsilon$ with $t \sim \{0, \dots, T-1\}$.

    Predict denoised sample $\hat{y} = f_\theta(x_t, t, c)$ and reconstruct $x_0^{\text{pred}}$.

    Update $(E_\phi, f_\theta)$ using diffusion loss (denoising + smoothness + clearance).

    In the environment, propose candidate set $\mathcal{C}_t$ via diffusion model.

    Sample action $A(t) \sim \pi_\psi(\cdot | S(t))$, execute sub-path, and obtain reward $R(t)$ and next state $S(t+1)$.

    Store transition $(S(t), A(t), R(t), S(t+1))$ in replay buffer $\mathcal{B}$.

    Update critics by minimizing Bellman error, update policy via entropy-regularized objective, and adjust temperature $\alpha$.

   **end for**

  **end for**

  **Inference** Given $(s, g, \mathcal{O})$, the diffusion model generates candidate set $\mathcal{C}_t$. The DRL agent selects $a^* = \arg\max_a Q(s, a)$ or samples stochastically from $\pi_\psi$, executes the sub-path, and repeats online replanning until termination.

---

TRAINING AND PLANNING HYPERPARAMETERS

DIFFUSION MODEL – TRAINING AND SAMPLING

Table 2: Diffusion model hyperparameters for training and sampling.

| Category | Parameter | Value |
|---|---|---|
| Data / schedule | Epochs / Batch | 500 / 64 |
| | Diffusion steps ($T_{\text{diff}}$) | 1000 |
| Geometry | Global horizon steps ($T$) | 64 |
| | Grid resolution ($H, W$) | $32 \times 32$ |
| | Segment-wise planning | On (fixed segments) |
| | Segment length / stride | 16, 32 / 4 |
| Network | Denoiser backbone | Transformer |
| | $d_{\text{model}}$ / heads / layers | 256 / 4 / 6 |
| | FFN / Dropout | 512 / 0.1 |
| | Conditioning dim | 256 |
| | Prediction parameterization | $v$-prediction |
| | Classifier-free dropout | 0.15 |
| Training-time proposal | Candidates per step ($K$) | 12 |
| | Clearance weighting $w_{\text{clear}}$ / margin | 0.5 / 0.02 |
| Sampling (standalone) | DDIM steps / $\eta$ / CFG scale | 50 / 0.0 / 2.0 |

DEEP REINFORCEMENT LEARNING – TRAINING AND ONLINE PLANNING

Table 3: Deep reinforcement learning training and online planning hyperparameters using DDIM proposals.

| Category | Parameter | Value |
|---|---|---|
| Planning geometry | Segment length / Exec stride | 16, 32 / 4 |
| | Max planning iterations | 128 |
| Kinematics / safety | Cruise speed $v_{\text{cruise}}$ / time step $dt$ | 6.0 m/s / 0.2 s |
| | Goal radius / Collision radius | 2.0 m / 1.0 m |
| | Time penalty (per step) | 0.15 |
| DDIM (online) | Grid ($H, W$) | $64 \times 64$ |
| | DDIM steps / $\eta$ / CFG scale | 10 / 0.0 / 3.0 |
| | MC samples | 1 |
| | Motion cap $\delta_{\text{max}}$ / line-blend | 0.12 / 0.01 |
| | Regularizers ($\lambda_{\text{len}}, \lambda_{\text{col}}$) | (1.0, 100.0) |
| Environment | Dynamic obstacles | 10 |
| SAC training | Episodes | 1000 |
| | Discount $\gamma$ / Target update $\tau$ | 0.99 / 0.005 |
| | Optimizer LR / Batch size | $3.0 \times 10^{-4}$ / 256 |
| | Replay capacity | 400,000 |
| | Warmup: start_steps / update_after | 2000 / 2000 |
| | Update_every / Eval cadence | 1 / every 20 episodes (10 eps) |
| | Actor/Critic MLP widths | [256, 256] |
| Reward shaping | $r_{\text{goal}}, r_{\text{collision}}, r_{\text{timeout}}$ | 1000,  500,  500 |
| | $r_{\text{prog}}, r_{\text{energy}}, r_{\text{uncert.}}, r_{\text{curv}}$ | 30.0,  25.0,  12.0,  7.0 |
| | $r_{\text{dist}}, r_{\text{clear}}$ | 0.4,  0.5 |

| Method | Success | Collision | Timeout | Energy | Clearance |
|--------|---------|-----------|---------|--------|-----------|
| Diffusion–RL (segment=16) | 0.92 | 0.00 | 0.08 | $348.8 \pm 23.2$ | 58.20 |
| Diffusion–RL (segment=32) | **1.00** | **0.00** | **0.00** | **392.9 $\pm$ 14.4** | 59.20 |
| Diffusion–RL (full horizon) | 0.52 | 0.00 | 0.48 | $168.9 \pm 27.6$ | 50.70 |
| Heuristic | 0.32 | 0.68 | 0.00 | $1065.8 \pm 12.4$ | 5.55 |
| Classical planner (RRT) | 0.37 | 0.63 | 0.00 | $505.1 \pm 47.8$ | 4.90 |
| Pure diffusion model | 0.27 | 0.01 | 0.72 | $406.1 \pm 38.4$ | 53.00 |
| Diffuser | 0.00 | 1.00 | 0.00 | $2417.0 \pm 74.6$ | 3.97 |
| MPD | 0.00 | 1.00 | 0.00 | $2597.8 \pm 55.4$ | 3.94 |
| MMD | 0.00 | 1.00 | 0.00 | $2580.0 \pm 61.4$ | 3.94 |
| SMD | 0.00 | 1.00 | 0.00 | $2643.8 \pm 58.6$ | 3.92 |
| MCTD | 0.00 | 0.00 | 1.00 | $280.9 \pm 17.1$ | 26.07 |
| Stitch | 0.00 | 1.00 | 0.00 | $4917.9 \pm 395.2$ | 3.93 |

EXPERIMENTAL RESULTS

