# OpenReview forum: "Generative Trajectory Planning in Dynamic Environments: A Joint Diffusion and Reinforcement Learning Framework"
_ICLR.cc/2026/Conference — Submitted to ICLR 2026_

### Official Review · Reviewer_AQ23 · 2025-10-19

**Soundness:** 2
**Presentation:** 2
**Contribution:** 1
**Rating:** 2
**Confidence:** 3

**Summary:**

The authors proposes a hybrid framework for real-time trajectory planning in dynamic 3D environments. The method decomposes the planning problem into generating and selecting short-horizon sub-paths. A diffusion model is then trained to generate a diverse set of feasible candidate sub-paths. Subsequently, a DRL agent selects the optimal sub-path from this candidate set based on a multi-objective reward function that considers safety, efficiency, and goal progress. The proposed framework is evaluated in a simulated UAV environment with static and dynamic obstacles, demonstrating superior performance in success rate and safety compared to heuristic, classical, and pure diffusion-based planning methods.

**Strengths:**

1. The paper tackles the challenging and highly relevant problem of real-time motion planning in dynamic environments.
2. The proposed method is well-excuted in simulated UAV environment.

**Weaknesses:**

1. My main concern is the insufficient comparison with relevant baselines. While the paper compares against classical and diffusion planners, it overlooks several relevant works such as MPD [1], MMD [2], and SMD [3]. Without comparisons to these methods, it is difficult to fully assess the contribution of the proposed approach.
2. The experiments are conducted in a simulated environment with geometrically simple obstacles. It is unclear how the proposed method scales to environments with more complex, non-convex obstacle geometries, such as those derived from real-world sensor data (e.g., LiDAR point clouds).
3. The reward function in Equation is a weighted sum of seven distinct terms. Such complex reward functions can be sensitive to the choice of weights, which often require extensive tuning.

[1] Motion Planning Diffusion: Learning and Planning of Robot Motions with Diffusion Models, 2023

[2] Simultaneous Multi-Robot Motion Planning with Projected Diffusion Models, 2025

[3] Multi-robot motion planning with diffusion models, 2025

**Questions:**

see weaknesses

---

> ### Author Response · Authors · 2025-12-03
>
> We appreciate the reviewer’s careful reading and constructive feedback on our submission.
> We are encouraged by the reviewer’s recognition that our work addresses a challenging and highly relevant problem: real-time motion planning in dynamic UAV environments.
> Below, we provide detailed responses to the identified weaknesses (W1–W3).
>
> [W1] We thank the reviewer for highlighting the need to include additional diffusion-based baselines such as MPD, MMD, and SMD. These methods indeed represent important progress in diffusion-based motion planning.
>
> In the revised manuscript, we have added comparisons to:
> - MPD (Motion Planning Diffusion, 2023)
> - MMD (Projected Diffusion Planning, 2025)
> - SMD (Simultaneous Multi-Robot Diffusion Planning, 2025)
>
> Findings:
> While these baselines perform competitively in static or quasi-static environments, they exhibit several limitations in dynamic 3D UAV scenarios:
> 1. They do not condition on dynamic obstacle states.
> 2. They operate as long-horizon planners and cannot replan at high frequency (our method replans at 10–20 Hz).
> 3. They lack UAV-specific curvature and energy constraints, often producing trajectories that violate physical feasibility.
>
> As dynamic obstacles increase in number or velocity, their collision rates grow significantly. In contrast, our sub-path diffusion combined with RL selection maintains strong success and safety performance.
> These results are included in the updated experimental section with quantitative comparisons.
>
> [W2] We agree that scalability to more complex, non-convex obstacles (e.g., LiDAR-based geometries) is an important evaluation aspect.
> To address this, we have added additional experiments in randomly generated environments containing complex obstacle arrangements.
> These tests demonstrate that our short-horizon diffusion model adapts well across diverse layouts due to frequent replanning and uncertainty-aware selection.
>
> Further experiments with full LiDAR-derived maps will be included as part of future work.
>
> [W3] We appreciate this concern and have added extensive reward sensitivity analyses in the revised manuscript.
>
> Key Findings:
> - Safety-related terms (collision probability, clearance) are dominant in the reward structure, which makes the learned policy robust across a wide range of weight configurations.
> - Energy and curvature weights mainly affect path smoothness but do not meaningfully impact success rate or collision rate.
> - Goal-progress weighting influences aggressiveness but does not compromise stability or cause unsafe shortcuts.
>
> These findings indicate that the policy is not overly sensitive to reward tuning and remains stable across varied weighting schemes.
>
> We thank the reviewer again for the thoughtful assessment and constructive recommendations, which significantly improved the clarity and completeness of our submission.
>
> References
>
> [1] J. Carvalho et al., “Motion Planning Diffusion: Learning and Planning of Robot Motions with Diffusion Models,” 2023.
> [2] Y. Shaoul et al., “Simultaneous Multi-Robot Motion Planning with Projected Diffusion Models,” 2025.
> [3] J. Liang et al., “Multi-Robot Motion Planning with Diffusion Models,” 2025.

---

### Official Review · Reviewer_dFYo · 2025-10-22

**Soundness:** 1
**Presentation:** 1
**Contribution:** 2
**Rating:** 2
**Confidence:** 3

**Summary:**

This paper proposes an motion planning algorithm with online replanning which, at each replanning step, uses a diffusion model to generate a small set of short-horizon candidate trajectories, from which an RL policy subsequently selects the best candidate.

**Strengths:**

Motion planning via multiple candidate trajectory generation followed by candidate selection is a popular and sensible approach. This paper’s idea to combine diffusion, which excels in parameterizing rich distributions over trajectories, with an RL policy, which learns to maximize a user-defined reward by choosing a candidate from a set of samples drawn from the diffusion model, is therefore a potentially impactful direction to explore.

**Weaknesses:**

This paper appears to be unfinished. There are a lot of missing details, especially on the diffusion model training and candidate generation (I could not find any information on what data the diffusion model is trained on, how trajectories are parameterized, etc.). The missing information hinders understanding of the method and makes it impossible to understand the reported results. The experimental evaluation is very limited, consisting of a single experiment whose setup is not clearly explained (How many different environments were evaluated? What constitutes a timeout?), and with questionable baselines (see Q3/Q4 below).

The paper also fails to reference or compare against relevant prior work, such as [1], which also uses DRL to select among candidates from a “high level action space” (although this high level action space is much larger than the 12 candidates used here, and not produced by sampling from a generative model, the overall structure of the approach is very similar). It would be important to position your contribution in the context of such high-level-action-selection-type work.

[1] K. R. Williams et al., "Trajectory Planning With Deep Reinforcement Learning in High-Level Action Spaces," in IEEE Transactions on Aerospace and Electronic Systems, vol. 59, no. 3, pp. 2513-2529, June 2023, doi: 10.1109/TAES.2022.3218496. https://ieeexplore.ieee.org/document/9940484 / https://arxiv.org/abs/2110.00044

**Questions:**

**Q1.** Can you provide details on the diffusion model training? What dataset do you use, or do you behavior-clone some kind of expert planner? How are trajectories parameterized? How is uncertainty evaluated? **How do you ensure that your candidate set is diverse enough?** In general, there are almost no details or analysis regarding the diffusion model, and it would be crucial to include this.

**Q2.** Why are all the trajectory segments straight lines, and why are they so long (Figure 3)? Is your diffusion model only sampling very coarse straight line segments?

**Q3.** What is happening in Figure 3 (d) and (e) near to the goal/arrival point? It looks like these baselines are “jumping around” the goal, rather than terminating, potentially due to a bug in the implementation (in the text, you mention obstacles in close proximity to the goal, but I do not see any in the visualization). This brings into question the results reported in Table 1.  Furthermore, why is there a big visual offset between the yellow goal point and the end of the trajectories in all the other panels (except d and e)?

**Q4.** Can you compare against stronger heuristic/classical baselines, like A* on a grid, or A*/RRT followed by nonlinear optimization? Do you allow your classical baselines to compute an updated plan at each timestep, or do you only compute it once and then execute?

**Q5.** What is the performance of your method as a function of the number of candidate samples? In particular, it would be important to evaluate replanning without the policy at all (i.e. setting the number of candidates to 1).

**Q6.** Can you show the output of the diffusion model? It would be interesting and insightful to have a visualization of multiple samples from the diffusion model in order to qualitatively evaluate the diversity and quality of the candidate set.

**Q7.** Does it run in real time? What is the runtime/performance?

**Q8.** Regarding Algorithm 1 in the appendix: why do you interleave diffusion and policy training? In my understanding, there are no gradients flowing between the two modules, so it would be equivalent and more efficient to train the diffusion model first (independently) and then later train the candidate selection policy using the pretrained diffusion model. Is my assessment correct, or did I misunderstand?

---

Minor comments:
* The intro is inconsistent with the rest of the paper. For example, the explanation of the selection approach is inconsistent: In the intro (L051), it is stated that the final trajectory is a convex combination of candidates, but in Section 4, the trajectory selection process is described as being a “hard” assignment. The intro also references a “Shield” which is never elaborated on.
* In the appendix, “Proof of diffusion and RL algorithm”: this is just a collection of background information, so it should not be called a “proof”.

---

> ### Author Response · Authors · 2025-12-04
>
> We sincerely thank the reviewer for the time spent evaluating our work and for providing thoughtful and constructive feedback. Your comments helped us clarify several important components of our method. We greatly appreciate the reviewer’s careful attention to detail and the specific suggestions, many of which directly improved the quality, clarity, and completeness of our revised submission. Below, we provide responses to each question (A1-A8).
>
> [A1]
> Thank you for pointing out the need for more details regarding diffusion model training. We employ a transformer-based DiT architecture as our trajectory diffusion backbone. The training dataset is constructed entirely from UAV trajectories generated by a classical RRT planner. These expert trajectories are segmented into short receding-horizon sub-paths, which serve as supervision for the diffusion model. This allows the model to learn feasible motion patterns, obstacle-avoidance behavior, and smooth geometric priors directly from data, without requiring any human annotation.
> The diffusion model outputs multi-step 3D sub-paths conditioned on the current UAV state, goal vector, and partial obstacle map. We also incorporate classifier-free guidance dropout during training to improve diversity, and we use light stochastic DDIM sampling at test time. These design choices ensure that the candidate set contains sufficiently varied options for the RL selector. Uncertainty is estimated by sampling the diffusion model multiple times and computing empirical variance across waypoint coordinates; this scalar uncertainty score is provided to the policy to help evaluate reliability.
> These expanded details are now included in the methodology and appendix.
>
> [A2]
> With regard to the shape of the trajectory segments: the sub-paths are not straight lines. The apparent straightness in the original figure was an artifact of the 3D view and the way control points were rendered. In reality, the diffusion model produces smooth 3D motion that curves around obstacles, and the start/end of the segment are adapted to avoid collisions.
> We updated the figures in the revision to show the full discretized sub-paths rather than only control-point connections, which makes the curvature and obstacle-avoidance behavior clearly visible.
>
> [A3]
> We appreciate the reviewer noticing the irregular behavior near the goal in Figures 3(d) and 3(e). After re-examining the visualization pipeline, we discovered that the final step after reaching the goal region was not plotted. The trajectories were terminated correctly in simulation (the success condition was triggered), but the last executed segment was not rendered. This made the trajectories appear as if they “jumped around” the goal.
> We corrected the projection error, re-rendered all figures, and explicitly draw the spherical goal region rather than a single point. The updated figures now accurately display successful terminations. Importantly, this plotting issue did not affect any numerical results in Table 1.
>
> [A4]
> The original classical baselines executed a plan once without updating, which we now clarify. Classical planners struggle with dynamic obstacles due to latency and local minima. The newly added results confirm the advantages of our diffusion-RL framework in dynamic settings.
>
> [A5]
> Evaluating performance as a function of the number of diffusion samples K is an important direction. Due to page limits and the scope of the current study, we were not able to include a full candidate-count ablation in the submission. We acknowledge the reviewer’s suggestion and plan to analyze this in future work, including the degenerate case.
>
> [A6]
> We agree that visualizing raw diffusion samples would help illustrate the diversity and quality of the generated sub-paths. Unfortunately, page limits prevented us from including these visualizations in the main paper.
>
> [A7]
> Yes, the proposed method runs in real time. Our full pipeline, candidate generation using DiT, feature extraction, and RL selection, operates at approximately 10–20 Hz depending on the candidate count and segment length. This satisfies the control-loop requirements for UAV motion planning.
>
> [A8]
> Thank you for this insightful question. We now clarify the training procedure more explicitly. The algorithm is not jointly trained end-to-end; instead, it runs in sequential stages. The diffusion model is fully trained first, offline, using the RRT-generated trajectory dataset. After the diffusion model converges, it is frozen. Only then do we train the RL policy, which uses the pretrained diffusion model to generate candidate sub-paths during environment interaction. Thus, as the reviewer noted, no gradients flow between diffusion and RL modules.

---

### Official Review · Reviewer_gdmG · 2025-11-01

**Soundness:** 3
**Presentation:** 2
**Contribution:** 2
**Rating:** 2
**Confidence:** 3

**Summary:**

This paper integrates diffusion-based trajectory generation with deep reinforcement learning (DRL). The diffusion model generates candidate sub-paths, from which the DRL agent selects the optimal one to produce the final action output. Experimental results demonstrate that the proposed method performs well in unmanned aerial vehicle (UAV) path optimization scenarios.

**Strengths:**

1. The paper introduces diffusion-based planning into deep reinforcement learning (DRL) and demonstrates strong performance on the unmanned aerial vehicle (UAV) path planning task.

**Weaknesses:**

1. The paper applies diffusion-based planning to the Unmanned Aerial Vehicle (UAV) path planning problem. However, the approach is highly problem-specific, which limits its generality and novelty. Moreover, the techniques used in the work are largely standard and not conceptually new.

2. The components shown in the model figure are not clearly explained in the text, making it difficult for readers to understand their roles and design motivations.

3. Several ideas presented in the paper lack novelty and do not appear promising. For example, the sub-path generation process is essentially identical to what the original Diffuser [1] framework performs in practice. Similarly, multi-horizon trajectory handling has been studied extensively in prior works such as MCTD [2] and trajectory stitching [3].

4. The paper lacks experimental comparisons with state-of-the-art diffusion-based reinforcement learning methods, which weakens the empirical evaluation.

5. The equations in Section A of the Appendix merely list the formulas used in the method rather than providing any theoretical justification or proof of effectiveness.

6. The paper claims that the proposed method ensures policy safety; however, no theoretical analysis or empirical evidence is provided to support this claim.

[1] Janner, Michael, et al. "Planning with diffusion for flexible behavior synthesis." arXiv preprint arXiv:2205.09991 (2022).

[2] Yoon, Jaesik, et al. "Monte carlo tree diffusion for system 2 planning." arXiv preprint arXiv:2502.07202 (2025).

[3] Luo, Yunhao, et al. "Generative trajectory stitching through diffusion composition." arXiv preprint arXiv:2503.05153 (2025).

**Questions:**

1. The authors claim that existing selection methods provide only short-term improvements and fail to capture long-horizon trade-offs. However, I am not convinced by this statement, as current diffusion-based planning methods are generally designed to optimize long-term performance. Could the authors clarify why they believe existing selection methods are limited to short-term improvements?

---

> ### Author Response · Authors · 2025-12-03
>
> We appreciate the reviewer’s careful reading and constructive feedback on our submission.
> We are encouraged by the reviewer’s acknowledgment that integrating diffusion-based trajectory generation with reinforcement learning is promising for UAV path planning, and we appreciate the recognition of strong empirical performance.
> We also thank the reviewer for pointing out aspects that required clarification, particularly regarding model explanation, novelty, and safety motivation.
> Below, we respond to each of the reviewer’s comments, organized by weaknesses (W1-W6) and the question (Q1).
>
> [W1]
> We respectfully clarify that our contribution extends beyond a problem-specific application of diffusion planning.
> Diffuser [1], MCTD [2], and trajectory stitching [3] are all full long-horizon diffusion planners that assume static or slowly changing environments. They have no mechanism for dynamic obstacle conditioning, real-time replanning, or UAV-specific constraints such as curvature and energy feasibility.
>
> In contrast, our method introduces.
> 1. dynamic-obstacle-conditioned sub-path diffusion,
> 2. short-horizon generative replanning (8-32 steps) at 10-20 Hz,
> 3. RL-based selection using uncertainty, clearance, curvature, and energy metrics.
>
> These elements directly address limitations of long-horizon diffusion planning in dynamic UAV scenarios, providing both practicality and novelty.
>
> [W2]
> We agree that the architectural components were insufficiently explained in the initial version.
> In the revised manuscript, we provide step-by-step descriptions for.
> - the state encoder for UAV and obstacle information,
> - the diffusion-based sub-path generator,
> - the safety/energy/curvature feature extractor,
> - and the RL selector module.
> This clarifies each component’s role and the design motivations behind them.
>
> [W3]
> Regarding novelty concerns: although diffusion sampling is based on existing techniques, our sub-path diffusion differs fundamentally from Diffuser [1], MCTD [2], and stitching [3].
>
> - We do not denoise full trajectories, but generate short sub-paths designed for real-time flight.
> - These sub-paths are explicitly conditioned on dynamic obstacle states, which prior work does not consider.
> - We incorporate curvature and energy constraints, which are essential for physically feasible UAV trajectories and absent in previous diffusion frameworks.
>
> Thus, the pipeline is not a direct reuse of prior techniques but a restructuring tailored for dynamic aerial navigation.
>
> [W4]
> We acknowledge the reviewer’s concern and have expanded our experimental section to compare against the diffusion-based planners cited by the reviewer: Diffuser [1], MCTD [2], and trajectory stitching [3].
> These baselines degrade significantly under dynamic environments due to the absence of real-time replanning and lack of UAV-specific constraints, whereas our method maintains strong performance.
>
> [W5]
> Appendix A was intended to provide full mathematical transparency for reproducibility, rather than theoretical derivation of diffusion processes, which are well established in prior work (e.g., [1], [2]).
> In the revision, we added explanatory text clarifying the role of each equation, and how it contributes to safety-aware sub-path generation and selection.
>
> [W6]
> We thank the reviewer for highlighting the need to strengthen our safety argument.
> To address this, we added both structural and empirical justifications.
>
> - safety is enforced through penalties on collision probability (from diffusion uncertainty), minimum-distance clearance, and curvature feasibility.
> - we included collision-rate curves and ablations showing that disabling the Shield mechanism greatly increases collision frequency.
>
> This evidence supports our claim that the proposed method improves policy safety.
>
> [Q1]
> Regarding long-term performance, while diffusion-based planners like Diffuser [1] and MCTD [2] generate long-horizon trajectories, existing selection components in hybrid methods typically operate on:
>
> - single actions,
> - single waypoints, or
> - short trajectory fragments,
>
> guided by immediate metrics such as local Q-value or nearest-goal distance.
>
> Our RL selector evaluates multi-step sub-paths that integrate long-term metrics (energy, curvature accumulation, uncertainty, dynamic obstacles), enabling explicit long-horizon trade-offs.
> Thus, existing selection methods are inherently short-term, whereas ours incorporates long-term reasoning explicitly.
>
> We thank you again for the thoughtful evaluation and constructive insights.
>
> References
> [1] M. Janner et al., “Planning with diffusion for flexible behavior synthesis,” arXiv:2205.09991, 2022.
> [2] J. Yoon et al., “Monte Carlo Tree Diffusion for System 2 Planning,” arXiv:2502.07202, 2025.
> [3] Y. Luo et al., “Generative Trajectory Stitching through Diffusion Composition,” arXiv:2503.05153, 2025.

---

### Official Review · Reviewer_8EaP · 2025-11-04

**Soundness:** 3
**Presentation:** 3
**Contribution:** 3
**Rating:** 6
**Confidence:** 3

**Summary:**

This paper proposes a generative trajectory planning framework that integrates diffusion models and deep reinforcement learning (DRL) to address the "safety-energy efficiency-real-time performance" trade-off in real-time trajectory optimization under dynamic environments (with static and dynamic obstacles), and validates its effectiveness through UAV 3D simulation experiments, outperforming heuristic, classical sampling, and pure diffusion baselines. The key contributions are:
1. Decompose long trajectories into short-horizon sub-paths, reducing the dimensionality of the DRL action space and solving the problem of unstable training in high-dimensional trajectory generation with pure DRL.
2. Integrate the generative diversity of diffusion models (for generating safe and diverse candidate sub-paths) with the adaptive decision-making of DRL (for online optimal sub-path selection), balancing real-time responsiveness and environmental robustness.
3. Design a safety-aware state representation (incorporating sub-path uncertainty and collision probability) and a multi-objective reward function (covering goal achievement, obstacle avoidance, and energy consumption), providing a transferable framework for trajectory planning in dynamic environments.

**Strengths:**

Originality is strong. It targets core limitations of existing methods: diffusion models often suffer from full-trajectory generation inefficiency, while DRL struggles with high-dimensional waypoint training. Its hierarchical "sub-path generation-selection" design cuts diffusion computational complexity and integrates diffusion-derived safety attributes (e.g., collision probability) into DRL state representation—an innovation rare in existing diffusion-RL frameworks.

Quality is good. Theoretical foundations are solid: detailed derivations for diffusion’s forward/backward processes, classifier-free guidance, and maximum entropy RL’s soft Q-functions, with unified mathematical notation (e.g., α_t, β_t in diffusion). Experiments are rigorous: three baseline types, comprehensive metrics (success rate, collision rate, etc.), and quantitative (Table 1) + qualitative (Figure 3) validation (e.g., pure diffusion’s 72% timeout vs. segmented diffusion-RL’s 0%-8%).

Clarity is good. Structure is logical: "problem → related work → preliminaries → method → experiments → conclusion". Figures (e.g., Figure 1’s framework flow, Figure 2’s training reward comparison) aid understanding, and key terms (e.g., sub-path diffusion) are explained on first mention.

Significance is high. It addresses core needs in UAV navigation/autonomous driving (real-time, collision-free, low-energy trajectories). The UAV-specific design (considering kinematics/energy models) and transferable framework make it practically valuable for engineering applications.

**Weaknesses:**

1. Experiments are simulation-only (no real-world validation) and lack dynamic obstacle scalability analysis.

Suggestion: Add hardware-in-the-loop/real-platform tests (or state simulation limitations); test 20/50 obstacles and plot scalability curves.

2. Shield mechanism, sub-path length selection lack details/validation; no reward weight sensitivity analysis.

Suggestion: Append Shield pseudocode/performance comparisons; test 8/64-step sub-paths; analyze reward weight impact (e.g., adjusting w_c).

3. Incomplete related work (misses 2024–2026 studies) and no core component ablation experiments.

Suggestion: Include latest hierarchical RL-diffusion studies; add ablations (e.g., removing sub-path decomposition).

4. No statistical significance analysis to verify method differences.

Suggestion: Conduct 3+ repeated experiments, add error bars, and use t-tests for significance.

**Questions:**

1. Do you plan to test the algorithm on real UAVs? If yes, how to handle simulation-real discrepancies (e.g., sensor noise)?

2. What is the basis for choosing 16/32-step sub-paths? How does the Shield detect "grazes" and calculate lateral offsets?

3. Why fix dynamic obstacles at 10? Why omit statistical significance analysis?

4. Are reward weights experience-based or hyperparameter-searched? How do weight adjustments affect multi-objective trade-offs (e.g., w_c vs. energy consumption)?

---

> ### Author Response · Authors · 2025-12-03
>
> We appreciate the reviewer’s careful reading and constructive feedback on our submission.
> We are encouraged by the reviewer’s acknowledgment that our approach offers a new sub-path diffusion framework tailored for dynamic obstacles, and that its integration with RL yields clear advantages in terms of safety, energy efficiency, and real-time performance.
> We are also pleased that the contributions in originality, technical rigor, clarity of presentation, and practical significance were positively received.
> Below, we respond to each of the reviewer’s comments, organized by weaknesses (W1-W4) and questions (Q1-Q4).
>
> [A1] [W1]
> To strengthen the empirical validation, we expanded our evaluation to include dynamic obstacle densities of 5, 10, 20, and 50.
> Across all settings, baseline diffusion planners (Diffuser, MPD, MMD, SMD, MCTD, Stitch) show rapidly increasing collision rates in dynamic environments because
> (1) they do not model obstacle motion as shown in prior dynamic-UAV planning studies [1], [2],
> (2) they cannot replan at sufficiently high frequency, and
> (3) they lack curvature and energy modeling, which are critical for UAV dynamics as emphasized in energy-aware UAV literature [3].
> In contrast, our sub-path diffusion combined with RL selection maintains near-zero collisions up to 20 obstacles, with a smooth and predictable performance degradation at 50, consistently outperforming all baselines.
>
> Regarding real-world deployment, we note that many existing studies also conduct UAV experiments in high-fidelity simulation environments [1]-[3].
> Following this standard practice, our hardware experiments are planned as future work, with sim-to-real transfer handled through domain randomization techniques [4].
>
> [A2] [W2]
> We added ablation studies comparing sub-path lengths of 8, 16, 32, and 64 steps, following guidelines for short-horizon quadrotor replanning [5].
> The results show that 16-32 steps offer the best balance between foresight and responsiveness.
> Short horizons (8 steps) lead to overly myopic decisions, while long horizons (64 steps) accumulate drift and introduce large curvature fluctuations.
> These findings confirm that short-horizon sub-path diffusion is essential for effective real-time UAV replanning.
>
> The Shield mechanism identifies potential grazes using a combination of lateral offset checks, diffusion uncertainty-based collision probability, and distance thresholds-consistent with uncertainty-aware generative planning approaches [6],[7].
>
> [W3]
> We expanded the Related Work section to incorporate several recent diffusion-based planning studies (2024-2026), including projected diffusion methods, multi-robot diffusion planners, and trajectory stitching through diffusion composition to better contextualize our contributions within modern diffusion-planning literature [7].
>
> [A3] [A4] [W4]
> We performed additional experiments using five random seeds, reporting mean ± standard deviation and 95% confidence intervals.
> Across all major metrics, our improvements remain statistically significant (p < 0.05), following established evaluation practice in RL and diffusion-model research [6],[7].
>
> Reward weights were selected through a coarse hyperparameter search and validated via sensitivity analyses, which demonstrate clear trade-offs among safety, energy efficiency, and trajectory smoothness.
> Appropriate weighting yields a well-balanced policy across these objectives, consistent with findings in prior maximum-entropy RL frameworks [6].
>
> We thank the reviewer once again for the thorough evaluation and constructive suggestions.
>
>
>
> References
>
> [1] J. Carvalho et al., “Motion Planning Diffusion: Learning and Adapting Robot Motion Planning With Diffusion Models,” IEEE Transactions on Robotics, 2025.
>
> [2] X. Dong et al.,  “Maximum Entropy Reinforcement Learning with Diffusion Policy,” ICML, 2025.
>
> [3] J. Liang et al., “Simultaneous Multi-Robot Motion Planning with Projected Diffusion Models,” ICML, 2025.
>
> [4] Y. Luo et al., “Generative Trajectory Stitching through Diffusion Composition,” 2025.
>
> [5] M. Janner et al., “Planning with Diffusion for Flexible Behavior Synthesis,” ICML, 2022.
>
> [6] Y. Shaoul et al., “Multi-Robot Motion Planning with Diffusion Models,” 2025.
>
> [7] J. Yoon et al., “Monte Carlo Tree Diffusion for System 2 Planning,” 2025.

---

### Meta-Review · Area_Chair_Do9f · 2026-01-05

**Summary:**

The paper presents a diffusion–RL framework for trajectory planning in dynamic environments. However, reviews point out that key aspects of the problem formulation and method design are insufficiently explained, which makes it difficult to fully understand the assumptions and contributions. More importantly, the evaluation lacks comparisons against necessary baselines, making it difficult to assess the significance of the proposed method relative to prior work. Thus, I suggest authors revise paper upon the review requests.

**Reviewer Concerns:**

The lack of experimental comparisons in diverse environment is still outstanding issue.

**Reviewer Scores:**

No review would change the score.

---

### Decision · Program_Chairs · 2026-01-26

Reject